# Process Improvements for Direct Reduced Iron Melting in the Electric Arc Furnace with Emphasis on Slag Operation

**Marcus Kirschen** [1],*, **Thomas Hay** [2] and **Thomas Echterhof** [2]

1   Thermal Process Engineering, University of Bayreuth, Universitätsstrasse 30, D-94557 Bayreuth, Germany
2   Department for Industrial Furnaces and Heat Engineering, RWTH Aachen University,
    D-52074 Aachen, Germany; hay@iob.rwth-aachen.de (T.H.); echterhof@iob.rwth-aachen.de (T.E.)
*   Correspondence: marcus.kirschen@uni-bayreuth.de

**Abstract:** Steelmaking based on direct reduced iron (DRI, and its compacted derivative hot briquetted iron, HBI) is an anticipated important global alternative to current steel production based on $FeO_x$ reduction in blast furnaces due to its lower specific $CO_2$ emission. The majority of DRI is melted and refined in the electric arc furnace with different process conditions compared to the melting of steel scrap due to its raw material composition being rather different. We provide data and analysis of slag composition of DRI charges vs. steel scrap charges for 16 industrial electric arc furnaces (EAFs). Suggestions for optimized slag operation and resulting process improvements of DRI melting in the EAF are given. A dynamic mass and energy model of the DRI melting in the EAF is introduced to illustrate the implications of the adapted slag operation on the EAF process with DRI charges.

**Keywords:** electric arc furnace; direct reduced iron; process model; process improvement

## 1. Introduction

The global steel industry is subject to significant changes in order to decrease its $CO_2$ emission representing the most important contribution within the production industry sector to global $CO_2$ emission, approximately 7%. Current steel productions routes are mainly (1) ore reduction in the blast furnace and steel refining in the basic oxygen converter (BF-BOF) and (2) melting of steel scrap in the electric arc furnace (EAF) with $CO_2$ emissions in the range 1.6–2.2 $t_{CO_2}/t_{LS}$ (BF-BOF) and 0.25–1.1 $t_{CO_2}/t_{LS}$ (EAF), respectively [1–3]. An established alternative to coal and coke-based reduction of iron ores in blast furnaces is the ore reduction by coal or reformed natural gas (CO, $H_2$) to direct reduced iron (DRI) or hot briquetted iron (HBI) in shaft furnaces [4–6], rotary kilns [7–10] or rotary hearth furnaces [11]. The reduction of solid pellets is mainly realized at temperatures below melting, 900–1100 °C. With increased energy input and adapted reactor design, however, tapping of liquid iron is also possible [12,13]. Specific $CO_2$ emission figures are in the range 0.5–0.7 $t_{CO_2}/t_{DRI}$ [14,15] due to the lower reactor temperature and natural gas-derived process gas as an energy source and ore reducing agent.

Global DRI/HBI production surpassed 108 million tons in 2019 [16], with a significant potential to replace coal-based iron making in blast furnaces globally. Iron ore reduction by hydrogen only has already been realized by applying reformed natural gas with shift reactor and $CO_2$ removal unit [5,17] but closed due to technical and commercial reasons. New concepts for carbon-free DRI production are available [18]. Generally, the total FeO content of ore grades for DRI production, $FeO_{tot} > 65\%$, is higher than typical ore grades applied to the BF, $FeO_{tot} < 65\%$. Today, the majority of DRI/HBI is melted with a varying share of steel scrap in EAFs with specific $CO_2$ emission figures in the range of 0.9 to 1.8 $t_{CO_2}/t_{LS}$ (including local $CO_2$ intensity of electrical energy for EAF) [19,20].

## 2. Process Characteristics of DRI/HBI Melting and Refining in the EAF

The EAF production characteristics with high shares of DRI/HBI are, however, rather different from conventional melting of steel scrap due to the remaining 4–7 wt % gangue oxides and 1.5–4.3 wt % carbon (Table 1). If high amounts of DRI are charged continuously to the EAF, different power programs and adapted additions of lime, dolomitic lime or doloma are required.

**Table 1.** Typical direct reduced iron (DRI) compositions (in wt %).

| Prod. Site | C | $Fe_{met}$ | MgO | CaO | $SiO_2$ | $Al_2O_3$ | Metallization [1] |
|:---:|:---:|:---:|:---:|:---:|:---:|:---:|:---:|
| A | 2.0 | 88.4 | 2.0 | 0.7 | 3.7 | 0.5 | 81.5 |
| B | 1.8 | 91.4 | 0.3 | 0.4 | 2.4 | 0.9 | 95.1 |
| C | 1.7 | 80.5 | 1.5 | n.a. | 3.1 | 0.2 | 91.4 |
| D [21] | 2.5 | 88.9 | 0.3 | 1.0 | 1.5 | 0.4 | 94.4 |
| E [17] | 4.3 | 87.3 [2] | | | 3.8 | | 96.0 |
| F [17] | 4.0 | 83.0 [2] | | | 6.2 | | 94.0 |

[1]: $x_{Fe}/(x_{Fe} + x_{FeO})$; [2]: DRI with high share of $Fe_3C$; n.a.: not available.

DRI and HBI are charged with steel scrap in varying amounts to the EAF, depending on local costs and availability. HBI is usually charged with the steel scrap by buckets to the EAF, requiring only minor adaptions to the EAF process at HBI shares up to 10%. Sidewall natural gas burners are of low importance for DRI melting as the maximum efficient energy transfer of gas burners is related to a solid scrap in the EAF. DRI is charged in cold or hot conditions to the EAF continuously via the 5th hole in the EAF roof, with charge weight portions from 50% to >95%. Depending on the particular EAF shell design, the remaining melt volume (hot heel) increases up to 30% of the total melt volume in order to facilitate the melting of the charged DRI (modern EAF shell designs exist even with a higher share of the hot heel). In these cases, the power programs must be adapted to continuous charging of material, long flat bath conditions, and increased input of lime and dololime for slag forming. The usual specific consumption figures of lime and dololime for scrap charged EAFs, 30–45 kg/$t_{LS}$, result in a specific amount of slag in the range of 70–100 kg/$t_{LS}$, according to the CaO mass balance (Equation (1) with $m_{DRI} = 0$). The total mass of slag per heat, $m_{slag}$, is determined by the CaO input with the slag formers (lime, dololime) and DRI neglecting a small loss of CaO to the off-gas system with EAF dust (Equation (1)):

$$x_{CaO,Lime} \cdot m_{Lime} + x_{CaO,Dololime} \cdot m_{Dololime} + x_{CaO, DRI} \cdot m_{DRI} = x_{CaO,Slag} \cdot m_{Slag}, \quad m_i \text{ in kg or kg/}t_{LS}, x_i \text{ in wt \%} \quad (1)$$

With increased lime and dololime input for DRI heats in order to operate at standard slag basicity ($x_{CaO}/x_{SiO_2}$) in the range from 1.8–2.1 for minimum corrosion of the refractory lining, the specific amount and volume of the process slag are significantly increased up to 140 kg/$t_{LS}$ (Equation (1)). The increased input of burned slag formers increases the electric energy demand accordingly by approx. 0.37–0.50 kWh/kg [22].

Continuous charging of raw materials provides EAF operation at flat bath conditions of a steel melt volume at a smoothly increasing level which benefits arc stability and control and decreased noise at the work floor. The absence of a scrap pile in the DRI-charged EAF, however, results in power programs with lower arc length, i.e., lower arc voltage, and slightly lower efficiency of energy transfer to the steel melt. Third, the metal yield of DRI charges is often lower, $m_{charged\ metal}/m_{tapped\ steel}$ = 87–92%, due to the oxide gangue in DRI compared with heats of 100% steel scrap with medium or high quality, 90–94%. As a result of these DRI-specific process conditions, the specific electrical energy demand of DRI heats is higher than for melting of steel scrap heats for carbon steel grades. The according melting time and tap-to-tap time of DRI heats is significantly longer than for scrap heats. EAF production characteristics of scrap heats vs. DRI heats are given in Table 2.

**Table 2.** Range of typical production parameters of electric arc furnace (EAF) charges with scrap and scrap/DRI, respectively, for low alloyed steel grades.

| Charges Based on | | 100% Scrap | 80–95% DRI |
|---|---|---|---|
| Share of DRI/HBI | (%) | 0–5 (HBI) | 60–95 (DRI) |
| Electric energy demand | (kWh/t) | 340–390 | 530–680 |
| Natural gas | ($m^3$/t) | 5–10 | 0–2 |
| Oxygen | ($m^3$/t) | 25–37 | 20–35 |
| Coal and carbon fines | (kg/t) | 2–9 | 8–17 |
| Slag former (lime, doloma, etc.) | (kg/t) | 23–35 | 27–60 |
| Tap temperature | (°C) | 1600–1635 | 1600–1635 |
| Tap-to-tap time | (min) | 50–60 | 60–100 |
| Metal yield | (%) | 90–94 | 87–92 |

## 3. Increased Mass and Energy Efficiency by Controlled EAF Slag Operation

### 3.1. Slag Analysis as Helpful Tool to Monitor, Control and Improve EAF Operation

Besides control of arc length and, occasionally, of analysis of power harmonics [23], frequent slag sampling and analysis has been implemented as an efficient process monitoring tool in order to operate efficiently at foaming slag conditions with an appropriate slag viscosity that require a certain control of slag composition at MgO saturation [24–26]. In steelmaking processes with high generation rates of CO gas, e.g., increased share of pig iron, hot metal or DRI, the control of the slag composition is less important for slag foaming but for controlling the corrosion of the MgO-based lining caused by strongly MgO-undersaturated slags [21,27,28].

Examples of average slag compositions of EAF heats based on steel scrap charges and on charges with steel scrap and > 50% DRI are given in Table 3. Figures 1 and 2 visualize the distributions of slag composition with respect to MgO saturation [26]. The product portfolio covered rebar and construction steel grades (14 EAFs) to special steel grades (2 EAFs). The applied raw materials were steel scrap (EAF 1 to 8) and blends of steel scrap with 50% to 100% DRI (EAF 9 to 16). The sizes of the EAFs ranged from 60 t to >200 t tap weight, located in nine countries worldwide. Slag samples were taken from the EAFs shortly before tapping and analyzed at the steel plant laboratory. Only mislabeled slag data, e.g., those from transport ladles (i.e., FeO < 10% and CaO > 45%) or from raw materials (e.g., lime, DRI), were excluded from the data sets.

**Table 3.** Average slag compositions at the tapping of EAFs charged with steel scrap only and charged with scrap and >50% DRI (slag compositions in wt %).

| Scrap | # | CaO | $SiO_2$ | FeO | MgO | $Al_2O_3$ | MnO | $Cr_2O_3$ | Total [1] | σ FeO | Basicity | |
|---|---|---|---|---|---|---|---|---|---|---|---|---|
| EAF 1 [1] | 422 | 26.1 | 16.7 | 29.5 | 10.5 | 8.4 | 5.6 | 1.6 | 99.6 | 4.1 | 1.6 [4] | 1.5 [5] |
| EAF 2 [1] | 359 | 31.1 | 11.6 | 28.1 | 10.6 | 5.4 | 5.0 | 1.1 | 94.4 | 4.6 | 2.7 [4] | 2.5 [5] |
| EAF 3 [1] | 1216 | 25.6 | 13.5 | 34.5 | 11.3 | 6.0 | 6.4 | 2.5 | 100.7 | 5.3 | 1.9 [4] | 1.9 [5] |
| EAF 4 [1] | 472 | 25.6 | 12.1 | 29.7 | 9.4 | 14.5 | 4.6 | 2.1 | 97.9 | 4.9 | 2.1 [4] | 1.3 [5] |
| EAF 5 [2] | 149 | 27.3 | 8.8 | 40.2 | 8.3 | 3.5 | 7.0 | 3.2 | 99.4 | 5.2 | 3.1 [4] | 2.9 [5] |
| EAF 6 [1] | 424 | 28.4 | 12.6 | 36.2 | 3.8 | 8.7 | 9.6 | n.a. | 99.8 | 3.9 | 2.3 [4] | 1.5 [5] |
| EAF 7 [1] | 202 | 30.0 | 14.5 | 34.5 | 10.8 | 4.5 | 1.9 | 0.6 | 98.3 | 4.5 | 2.1 [4] | 2.1 [5] |
| EAF 8 [1] | 858 | 36.1 | 15.7 | 25.0 | 9.3 | 10.3 | 0.7 | n.a. | 97.5 | 4.0 | 2.3 [4] | 1.7 [5] |
| DRI | | CaO | $SiO_2$ | FeO | MgO | $Al_2O_3$ | MnO | $TiO_2$ | Total [2] | σ FeO | Basicity | |
| EAF 9 [1] | 132 | 27.0 | 16.0 | 31.1 | 14.9 | 6.0 | 1.9 | 1.2 | 98.0 | 8.3 | 1.7 [4] | 1.9 [5] |
| EAF 10 [1] | 29 | 39.2 | 16.6 | 31.8 | 5.4 | 5.9 | 1.4 | n.a. | 100.3 | 6.9 | 2.4 [4] | 2.0 [5] |
| EAF 11 [1] | 325 | 28.5 | 19.4 | 33.9 | 9.7 | 3.2 | 0.2 | 3.7 | 99.3 | 4.9 | 1.5 [4] | 1.7 [5] |
| EAF 12 [1] | 203 | 36.9 | 17.4 | 30.1 | 7.7 | 5.1 | 0.9 | n.a. | 95.5 | 5.5 | 2.1 [4] | 2.0 [5] |
| EAF 13 [3] | 519 | 32.4 | 18.7 | 28.5 | 10.4 | 8.7 | 1.2 | n.a. | 99.8 | 4.7 | 1.7 [4] | 1.6 [5] |
| EAF 14 [1] | 19 | 40.5 | 17.7 | 21.7 | 9.4 | 6.7 | 1.9 | 0.8 | 99.2 | 9.6 | 2.3 [4] | 2.0 [5] |
| EAF 15 [1] | 918 | 30.5 | 21.0 | 26.3 | 11.3 | 5.2 | 1.2 | n.a. | 95.9 | 4.5 | 1.4 [4] | 1.6 [5] |
| EAF 16 [1] | 123 | 38.9 | 18.2 | 31.5 | 4.3 | 5.6 | 2.0 | n.a. | 100.4 | 6.2 | 2.1 [4] | 1.8 [5] |

[1]: rebar and construction steel grades, [2]: specialty steel grades, [3]: construction and specialty steel grades; #: number of slag data; [4]: $B_2 = CaO/SiO_2$; [5]: $B_4 = (CaO + MgO)/(SiO_2 + Al_2O_3)$, n.a.: not available, σ standard deviation.

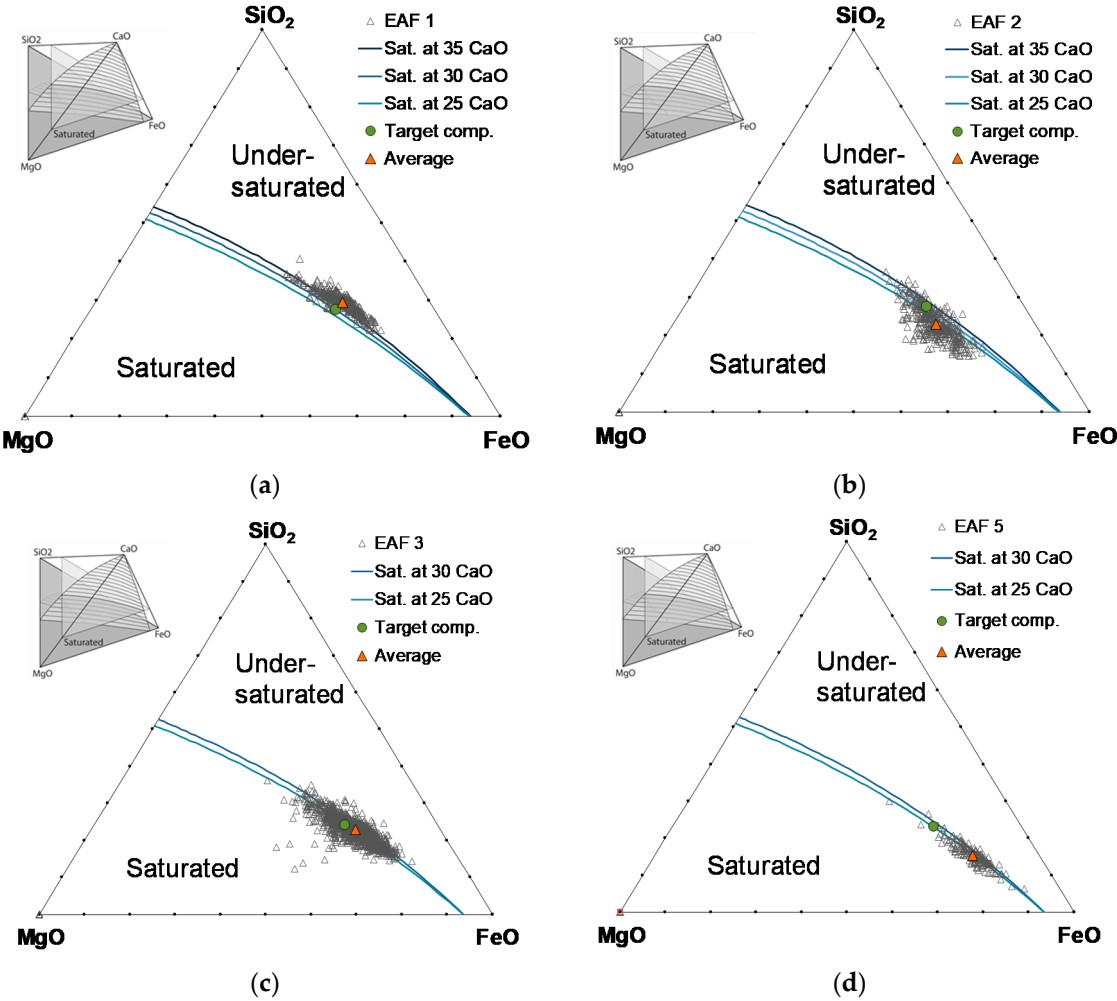

**Figure 1.** Distribution of slag analysis data from EAF heats for carbon steelmaking based on steel scrap (4 different EAFs (**a–d**)), in the system CaO-SiO$_2$-FeO-MgO-5% Al$_2$O$_3$; lines indicate MgO saturation at 25% to 35% CaO and 1600 °C [26].

Besides controlling the slag basicity, e.g., B$_2$ = x$_{CaO}$/x$_{SiO_2}$ or B$_4$ = (x$_{CaO}$ + x$_{MgO}$)/(x$_{SiO_2}$ + x$_{Al_2O_3}$), and MgO saturation, the control of FeO is important for avoiding unnecessary high Fe losses with the slag and increasing the metal yield; typical values are currently in the range of 25–30% FeO. Scatter of slag data provides information about the control on mass balance in the EAF, e.g., the appropriate input of CaO via lime and dololime for SiO$_2$ compensation and controlled slag basicity. Variations of FeO content of the tapped slag indicate the quality of FeO control by carbon and oxygen injectors. There is evidence from slag analysis data in Table 3 that (1) the average Fe loss in DRI-charged EAFs is higher than for steel scrap heats, and (2) the control of FeO is more difficult for DRI-charged EAF heats. The standard deviation of FeO is in the range of 3.9–5.5 for heats based on steel scrap and 4.1–9.6 for heats based on steel scrap, and >50% DRI. This is unexpected as DRI provides a more defined mass input to the EAF due to DRI production from iron ores and continuous monitoring of the DRI composition, in contrast to steel scrap with distinct quality classes and impurities. The lower control of slag compositions in DRI heats also seen in Figure 1 (heats based on steel scrap) vs. Figure 2 (heats based on >50% DRI).

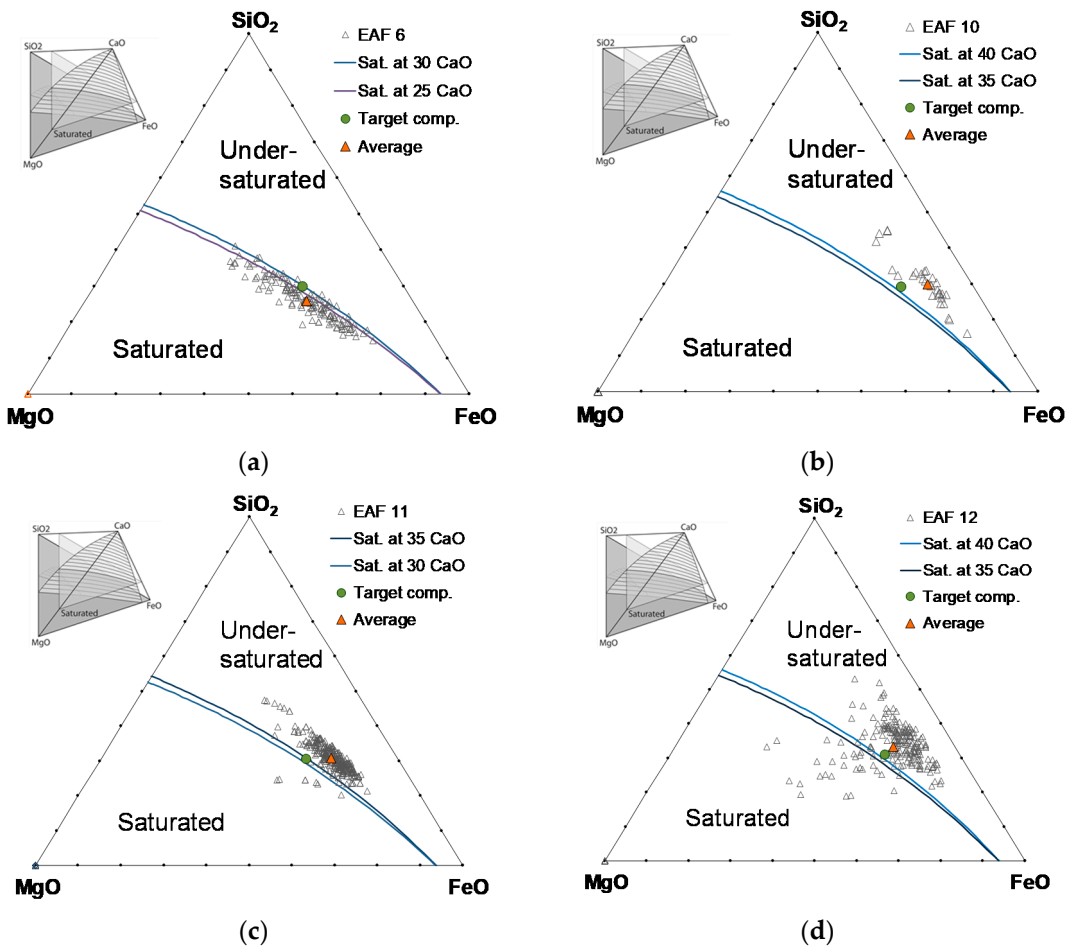

**Figure 2.** Distribution of slag analysis data from EAF heats for carbon steelmaking based on DRI (4 different EAFs (**a–d**)), in the system CaO-SiO$_2$-FeO-MgO-5% Al$_2$O$_3$; lines indicate MgO saturation at 25% to 40% CaO and 1600 °C.

*3.2. Slag Characteristics of DRI-EAF Heats*

The distribution of slag compositions for DRI heats is more complex than for heats based on steel scrap. In most cases, the EAF slags at tapping after melting and refining are close to the MgO saturation surface (Figure 1). This is due to the dissolution of MgO from the MgO-based EAF hearth and sidewall lining to the initial slag based on the remaining slag from the preceding heat and input of lime and dololime. Precisely defined mixtures of lime and doloma or another MgO-carrier provide an initial process slag closer to the MgO saturation with lower corrosion potential to the MgO hearth or MgO-C sidewalls (EAFs in Figure 1). Initial process slags based on lime or dolomitic lime only provide a higher corrosion potential (e.g., EAF 1 for scrap-based heats and EAF 11, and 12 for DRI-based heats in Table 3).

For DRI heats, the slag compositions show a more complex figure (Figure 2). The distribution and scatter of slag compositions at tapping are often higher than for scrap-based EAF heats. One reason is the EAF operation with an MgO-undersaturated initial slag if an MgO-carrier is not applied as slag former due to cost reasons or low availability of high-quality doloma or dolomitic lime. In these cases, the slag compositions show final MgO concentrations between the initial slag composition and the MgO saturation surface (e.g., EAF 10, 11, 12 in Figure 2). EAF 11 displays an unusually large distribution of MgO concentrations due to losses of MgO gunning and hearth repair mixes to the slag. Remarkably, the average slag composition of EAF 11 seems to be close to MgO saturation, although the slag operation is rather out of control.

Even more important for DRI-based heats is the control of FeO. Standard variation of FeO in Table 3 and the visual distribution of slag compositions in Figure 2 indicate a tendency to lower FeO control for DRI charges. Independent from basicity control, FeO ranges from 20 to >45%. Even if the lime and dololime input provides a more balanced process slag in order to operate at MgO saturated conditions (e.g., EAF 9 in Figure 2), the distribution of FeO at tapping is higher than usual. Recalling that composition control of DRI metallic raw material is usually better than for steel scrap due to better control on raw materials, continuous DRI production and usually DRI production on-site, low FeO control depends more likely on inappropriate EAF operation than on DRI input.

### 3.3. Slag Operation of DRI Heats at Lower Total Slag Mass

Specific slag masses are given in Figure 3, Equation (1), for heats based on steel scrap (up to 5% HBI) and DRI-based heats, assuming 2.5% $SiO_2$ and 0.5% CaO in DRI (Table 1) and 33% CaO in slag (Table 3). As the slag mass is higher for DRI heats due to the increased $SiO_2$ input, similar FeO concentration of the slag represents an elevated loss of Fe, approximately 0.9 $kg/t_{LS}$ Fe loss per% FeO in 120 kg/t slag (instead of 0.5 $kg/t_{LS}$ Fe loss per% FeO for steel scrap-based heats with lower slag mass, 70 kg/t). It is very likely that the lower control on FeO in the slag of DRI heats is due to the significantly increased slag volume that is continuously discharged from the EAF during the second half of DRI melting at elevated melt levels. Then, the retention time of the slag in the EAF is short, resulting in FeO levels with low control by carbon injectors or mixing with steel melt. With lower total slag volume, the discharging of slag starts at a later period of melting, providing an increased reduction of FeO by increased mixing with steel melt. Operating DRI heats at lower slag volume, however, requires lower input of lime, doloma, i.e., EAF operation at lower basicity in the range $x_{CaO}/x_{SiO_2}$ = 1.6–1.7 (Figure 3). In order to control and minimize corrosion of the MgO hearth and lining at low slag basicity, it is important to operate with slags near the MgO saturation from the very beginning by the adapted mass balance of lime and MgO carrier. MgO saturation can be efficiently monitored and controlled with diagrams in Figures 1 and 2 [26] or other saturation diagrams [23–25]. Some DRI-EAFs are operating at low basicity (EAF 9, 11, 13, 15 in Table 3), but half of the DRI-EAFs are still operating at high slag volumes and basicity near 2 (EAF 10, 12, 14, 16 in Table 3) following general EAF operating rules in order to minimize corrosion of the MgO lining and to operate at efficiently foaming slag in steel scrap charges. Due to increased input of carbon with DRI compared to steel scrap, CO formation is increased, and slag foaming is generally very well for DRI heats.

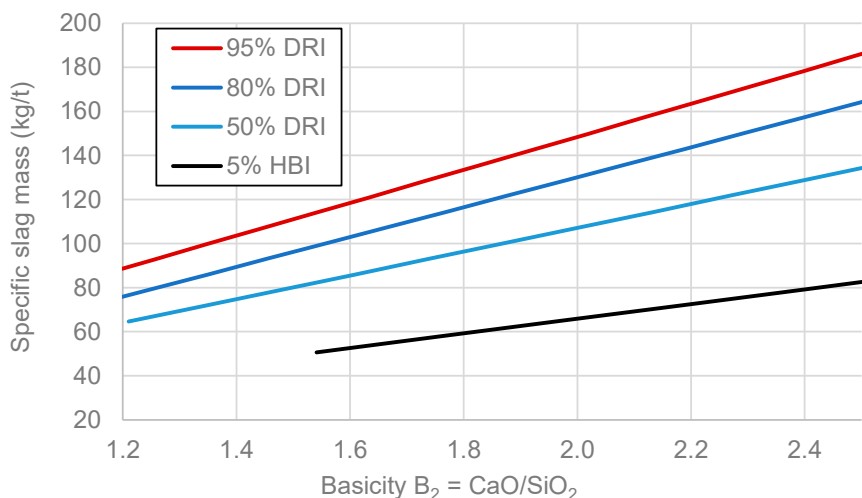

**Figure 3.** Specific slag volume as a function of slag basicity ($x_{CaO}/x_{SiO_2}$) for various ratios of DRI/hot briquetted iron (HBI) to steel scrap (assuming 2.5% $SiO_2$ and 0.5% CaO in DRI and 33% CaO in slag, Equation (1)).

## 4. EAF Process Modeling and Results.

The impact of a decreased use of slag formers on the energy efficiency of the EAF process was evaluated using a comprehensive process model for continuous charging of DRI developed from the EAF model for discontinuous charging of steel scrap proposed by Meier et al. [29–31] and described in previous publications [32,33].

### 4.1. Consistent Mass and Energy Balance Model for EAF Process

The model had to be adjusted for the DRI-based process since it has previously been developed and validated only for the scrap-based operation of EAF. The simulation was therefore adapted to accept a DRI feed rate instead of a scrap charge from a bucket. The DRI is added to the solid scrap zone in the model, and the parameters determining the heat transfer to the scrap zone were adjusted to account for the different behavior of the DRI compared to a pile of scrap melting down. Furthermore, while previously a $SiO_2$ and FeO content of the feed material could be set in addition to the Fe and trace elements, the significant gangue content of DRI necessitated the consideration of the CaO, MgO and $Al_2O_3$ content as well. The oxides are added to the liquid slag zone as the DRI melts. The option to define a mass flow of slag formers such as lime and doloma was already present in the model and could therefore be used without additional adjustments. The de-slagging, however, was added by defining a mass flow that is removed from the liquid slag zone and increases with the height of the slag (being zero below a set threshold). Other parts of the model, such as the heat transfer and chemical models, mainly remained unchanged as the operation with DRI is mostly identical to the flat-bath phase of the scrap-based process.

The operation chart determining inputs such as the electrical power or the mass flows of oxygen and carbon fines was based on the operation of an industrial EAF with a 100% DRI charge. Additional inputs, as well as empirical model parameters and the furnace geometry, were estimated based on similar sized EAF for scrap melting, for which extensive validation data were available. Based on this operation chart, the input of lime and dololime was both increased and decreased in 10% steps simulating an input from 70% to 150% of slag formers compared to the original industrial operation. The initial slag and hot heel composition were based on the average measured compositions available from the data, and the DRI composition was set according to measured values as well with a temperature of the charged DRI of 250 °C. Hot heel and initial slag mass and the beginning of the heat were estimated, and the parameters determining the de-slagging rate and the heating and melting rates of the DRI were adjusted so that all DRI is melted by the end of the heat and the slag amount remaining is similar to the initial slag mass as would be expected for the continuous operation of the furnace.

While the model has been extensively validated using process data from scrap-based operations [29–31,33] and remains mostly unchanged, the data available for this study was limited to the average operation chart and compositions already mentioned as well as tapping temperatures for three heats with the time of measurements given as within 3–5 min before tapping. Therefore, only the general temperature range, as well as qualitative results such as the already mentioned expected melting of all DRI and amount of slag produced and removed from the furnace, were available for validation of the model results. As the aim of this study is only a rough estimation and qualitative evaluation of the impact of the number of slag formers used in the process, this is, however, considered to be sufficient.

### 4.2. Results–Implications on Energy Balance, Savings and Productivity

As expected, the results showed the energy demand for the same tapping temperature increasing with a higher amount of slag formers charged. The adapted electrical energy input was adjusted by increasing or decreasing the power-on time of the same power program accordingly. Additionally, the basicity ratio changes as expected for the constant amount of DRI and therefore $SiO_2$ charged and varying amounts of CaO delivered from the added lime and doloma. The results of the simulations are summarized in Figure 4

concerning savings in power–on time and specific electrical energy demand by decreasing the lime and dololime input.

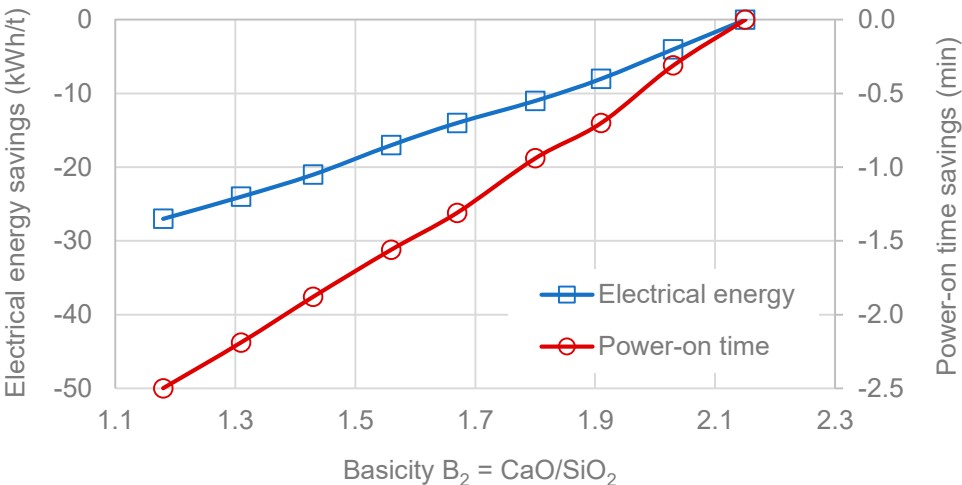

**Figure 4.** Calculated influence of lime and dololime input via slag basicity on power-in time and electrical energy demand of a 100% DRI-charged EAF model.

## 5. Discussion

Increased use of DRI in the EAF process increases the input of $SiO_2$ with DRI gangue oxides. An adapted EAF slag operation from standard basicity $x_{CaO}/x_{SiO_2} = 2.0$ to lower basicity values in the range 1.4 to 1.7 decreases the total slag mass and corresponding FeO loss to the slag as well as the electrical energy demand by approx. 8–17 kWh/t and consequently the power–on time. At lower slag basicity, however, the increased MgO saturation concentrations must be considered, e.g., by adapted saturation figures (Figure 2 [26]), in order not to increase the wear rate of the MgO lining. Due to the lower total slag mass, the efficiency of FeO reduction by carbon injection may be improved. In addition, the FeO losses by slag during DRI charging may be decreased: whereas dense HBI is usually charged with steel scrap by buckets to the EAF at minor amounts, DRI is charged continuously with or besides slag formers via the 5th hole in the EAF roof. Commercially produced DRI tends to form fines due to its high porosity [34]. Usually, the feeding spot is chosen near or between the electrodes (often between electrodes one and two near the slag door). Charged pellets fall into the steel melt due to the increased density of the reduced material (3.5–3.9 g/cm$^3$ [35]), whereas DRI fines stick to the slag layer above the steel melt.

Both the contamination of the slag samples with unreacted DRI fines and decreased efficiency of carbon injectors could be explanations of the observed increased FeO variation of slags from DRI charges in contrast to steel scrap charges (Table 3). An alternative DRI feeding spot between electrodes two and three (opposite to the slag door) could provide a longer retention time of DRI fines in the EAF for improved reduction of FeO residuals. Adapted oxygen injector positions may be necessary for an optimized DRI feeding spot.

The adapted EAF mass and energy model indicates differences between scrap charges and DRI charges due to different oxides input and flat bath conditions, e.g., improved slag foaming due to the carbon content of the DRI in agreement with common observation at DRI-EAFs. However, further improvements to the model are necessary in order to investigate the details of the DRI charging spot.

**Author Contributions:** Conceptualization, methodology, analysis of slag data, M.K.; modeling, software, validation, T.H., T.E.; writing, M.K., T.H., T.E.; All authors have read and agreed to the published version of the manuscript.

**Funding:** The APC was funded by the German Research Foundation (DFG) and the University of Bayreuth in the funding program Open Access Publishing.

**Conflicts of Interest:** The authors declare no conflict of interest.

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
