# Peer review of "Process Improvements for Direct Reduced Iron Melting in the Electric Arc Furnace with Emphasis on Slag Operation"

_processes, doi:10.3390/pr9020402_

Round 1

Reviewer 1 Report

The article is related to interesting topic that is currently actual in industry and has a great potential for the future. I see there only few minor week points that could be consider:

  1. The introduction is quite long, some mentioned data together with Table 1 could be part of another section.
  2. In article was found a different size of citation digit in Line 29 that needs to be unified - citation [1, 2, 3].
  3. Probably mistaken citation in Line 103. Regarding to Figure 1 and mentioned citation I suppose it meant to be reference source [26] instead of [23]. Same for the line 185.
  4. In Line 158 – probably wrongly typed „on“ instead of „of“ in the sentence part … „is the control on FeO.“

Author Response

Dear reviewer,

thank you for your supporting comments and careful reading of the manuscript. I corrected the typographical errors and updated the links to the correct references.

Table 1 with the corresponding paragraph was shifted to section 2.

Best regards, Marcus Kirschen

Reviewer 2 Report

The use of DRI (Direct Reduced Iron) or HBI (Hot Briquetted Iron) in steelmaking processes can become a more effective way in order to improve the process efficiency and environmental protection (especially on BOF processes, by reducing the usage of iron coming from blast furnace). However, the replacing of steel scrap with DRI (especially at EFA’s) it must to be studied/evaluated because of process efficiency in respect to metal yield.

The paper presents data and analysis of slag composition of DRI charges compared with steel scarp charges for 16 industrial available Electric Arc Furnaces. By comparing the results, the authors propose some improvements of DRI melting in EAF by considering the slag operation. In the paper a tentative dynamic mass and energy model for DRI melting in EAF is also proposed. However, this part should be better and more detailed presented.

However, the obtained results are valuable, containing a reasonable volume of information and can improve the knowledge in the field.

The paper must benefit of some improvements/revision:

  • The scientific paper could benefit of more added value if the study were conducted considering the use of DRI and steel scrap at each considered EAF’s;
  • In the references mentioned inside the text with [ ] please avoid the use of “e.g.”;
  • Try to zoom-in equation (1);
  • In references chapter, please use the Abbreviated Journal Name;
  • The discussion/conclusion chapter could benefit of a more detailed presentation, especially on the mass and energy model.

Author Response

Dear reviewer,

thank you for support and comments to improve the paper. Technical details on the references (e.g. and abbrevated journal names) are improved and corrected now in the revised version.

Concerning eq.1 I added the phrase: ... to the CaO mass balance (eq. 1 with mDRI = 0). The total mass of slag per heat, mslag, is determined by the CaO input with the slag formers (lime, dololime) and DRI neglecting a small loss of CaO to the off-gas system with EAF dust (eq. 1).

The section about the mass and energy model for an DRI-EAF will be enlarged and presented in more detail in an upcoming paper. I understand your point to get more details in this manuscript: 5 papers about the model are given as references, to save space and focus on the practical results here. I added a sentence to the conclusion section concerning the charging spot:

The adapted EAF mass and energy model indicates differences between scrap charges and DRI charges due to different oxides input and flat bath conditions, e.g. improved slag foaming due to the carbon content of the DRI in agreement with common observation at DRI-EAFs. However, further improvements of the model are necessary in order to investigate the details of the DRI charging spot.

Best regards, Marcus Kirschen